# Incidental temporal binding in rats: A novel behavioral task

**Dominika Radostova**[1,2], **Daniela Kuncicka**[1,2], **Branislav Krajcovic**[1,3], **Lukas Hejtmanek**[1], **Tomas Petrasek**[1,4], **Jan Svoboda**[1], **Ales Stuchlik**[1], **Hana Brozka**[1]*

**1** Institute of Physiology, Czech Academy of Sciences, Prague, Czechia, **2** Second Faculty of Medicine, Charles University, Prague, Czechia, **3** Third Faculty of Medicine, Charles University, Prague, Czechia, **4** National Institute of Mental Health, Klecany, Czechia

* hana.brozka@fgu.cas.cz

## Abstract

We designed a behavioral task called One-Trial Trace Escape Reaction (OTTER), in which rats incidentally associate two temporally discontinuous stimuli: a neutral acoustic cue (CS) with an aversive stimulus (US) which occurs two seconds later (CS-2s-US sequence). Rats are first habituated to two similar environmental contexts (A and B), each consisting of an interconnected dark and light chamber. Next, rats experience the CS-2s-US sequence in the dark chamber of one of the contexts (either A or B); the US is terminated immediately after a rat escapes into the light chamber. The CS-2s-US sequence is presented only once to ensure the incidental acquisition of the association. The recall is tested 24 h later when rats are presented with only the CS in the alternate context (B or A), and their behavioral response is observed. Our results show that 59% of the rats responded to the CS by escaping to the light chamber, although they experienced only one CS-2s-US pairing. The OTTER task offers a flexible high throughput tool to study memory acquired incidentally after a single experience. Incidental one-trial acquisition of association between temporally discontinuous events may be one of the essential components of episodic memory formation.

## Introduction

Temporal binding is the ability to associate successive non-overlapping events into a coherent sequence. Although the successive sub-events might be separated by time gaps, they might be experienced as a single memory [1]. Temporal binding is usually tested using trace association tasks, where two stimuli are separated by a time gap [2], making it a form of classical (Pavlovian) conditioning. Temporal binding can be a fundamental element necessary for creating complex memories [3], encoding the knowledge of personally experienced past events [4]. A key aspect of most natural memories is that they are acquired incidentally, i.e., events not intended to be memorized are remembered. The acquisition of these memories should not be dependent on conditioning or pre-training to the to-be-learned information. In an experimental setup, incidental memory can be tested when subjects are unaware that

**Data Availability Statement:** All data are available at: https://datadryad.org/stash/dataset/doi:10.5061%2Fdryad.q83bk3jn2 (DOI: 10.5061/dryad.q83bk3jn2).

**Funding:** This work was supported by Czech Science Foundation (GACR) grant 20-00939S awarded to A.S. www.gacr.cz. The funders had no role in the study design, data collection, analysis, decision to publish, or manuscript preparation.

**Competing interests:** The authors have declared that no competing interests exist.

they will be tested on recall [5–7]. To ensure that memory encoding is incidental, the memory test should be unexpected when the to-be-remembered event occurs [7]. Repeated test trials can lead animals to anticipate future memory testing. This repetition helps animals learn the task rules and identify relevant information, making them more motivated to encode the information intentionally. The evidence suggests that mechanisms of incidental memory acquisition might differ from acquisition with intent [8–11]. It has been proposed that one-trial memories are initially encoded as episodic, and that different memory systems might differ in their learning rates [12]. Our task featuring one-trial learning might prove helpful in testing this hypothesis.

To understand elements and prerequisites of complex memory, one of the vital tasks is to elucidate the acquisition and retrieval of events separated by a time gap [13–15]. Our goal was to broaden the palette of incidental and one-trial trace conditioning tasks by developing a novel and simple temporal binding task with a clear behavioral response and a balanced success ratio to enable discerning the underlying neurobiological changes in successful performers vs. non-performers. We focused on a one-trial design to ensure that memory was acquired incidentally. To achieve this, we took advantage of two natural behavioral tendencies of rodents: rodents avoid brightly lit environments [16] and rodents actively escape an immediate threat [17,18].

Our task, which we named One-Trial Trace Escape Reaction (OTTER), consists of three phases: habituation, pairing, and recall (Fig 1A–1C). The purpose of habituation is to familiarize rats with a novel environment and to reduce their exploratory activity. Each rat is habituated to two similarly constructed environmental contexts: an oval-shaped context A and a slightly larger rectangular-shaped context B (Fig 1A). Both contexts consist of one dark and one light chamber and are separated by a partition with a rectangular opening. This design allows us to exploit the natural tendency of rodents to avoid bright light. Even if rats are free to move between both chambers, they strongly prefer the dark chamber. To olfactorily distinguish both contexts, context A is cleaned with an alcohol-based wash, while context B is cleaned using a vinegar-based wash.

During the pairing session, rats experience two novel stimuli separated by a time gap (Fig 1B). Each rat is first allowed to explore the apparatus of context A or B exactly as it would during the habituation sessions. When the rat is settled in the dark chamber, it first hears a three-second acoustic cue (the conditioned stimulus CS), then receives an electric foot shock (unconditioned stimulus, US) two seconds after the CS stops. The US is terminated immediately after the rat escapes to the light chamber. This is the opportunity for the rat to incidentally associate the CS with the US (CS-2s-US) and learn that escape to the light chamber provides safety from the US.

The association between the CS and US is tested during the recall session 24 hours later (Fig 1C). Unlike in traditional active avoidance tasks, the recall in OTTER is tested in a different environmental context. Testing the recall in a different environmental context renders the association between the US and environmental context irrelevant; fear-related behavioral responses are therefore only attributable to the association between the CS and US. The recall session begins by placing the rat in the dark chamber of the environmental context other than the context of the pairing session (B or A). If at least 15 minutes elapsed and the rat rests in the dark chamber, the CS is delivered, and the rat's reaction is observed. There are two possible reactions: either the rat escapes into the light chamber ('responder') or remains in the dark chamber ('non-responder'). Here we present normative data from the OTTER task and discuss its strengths, limitations, and possible applications.

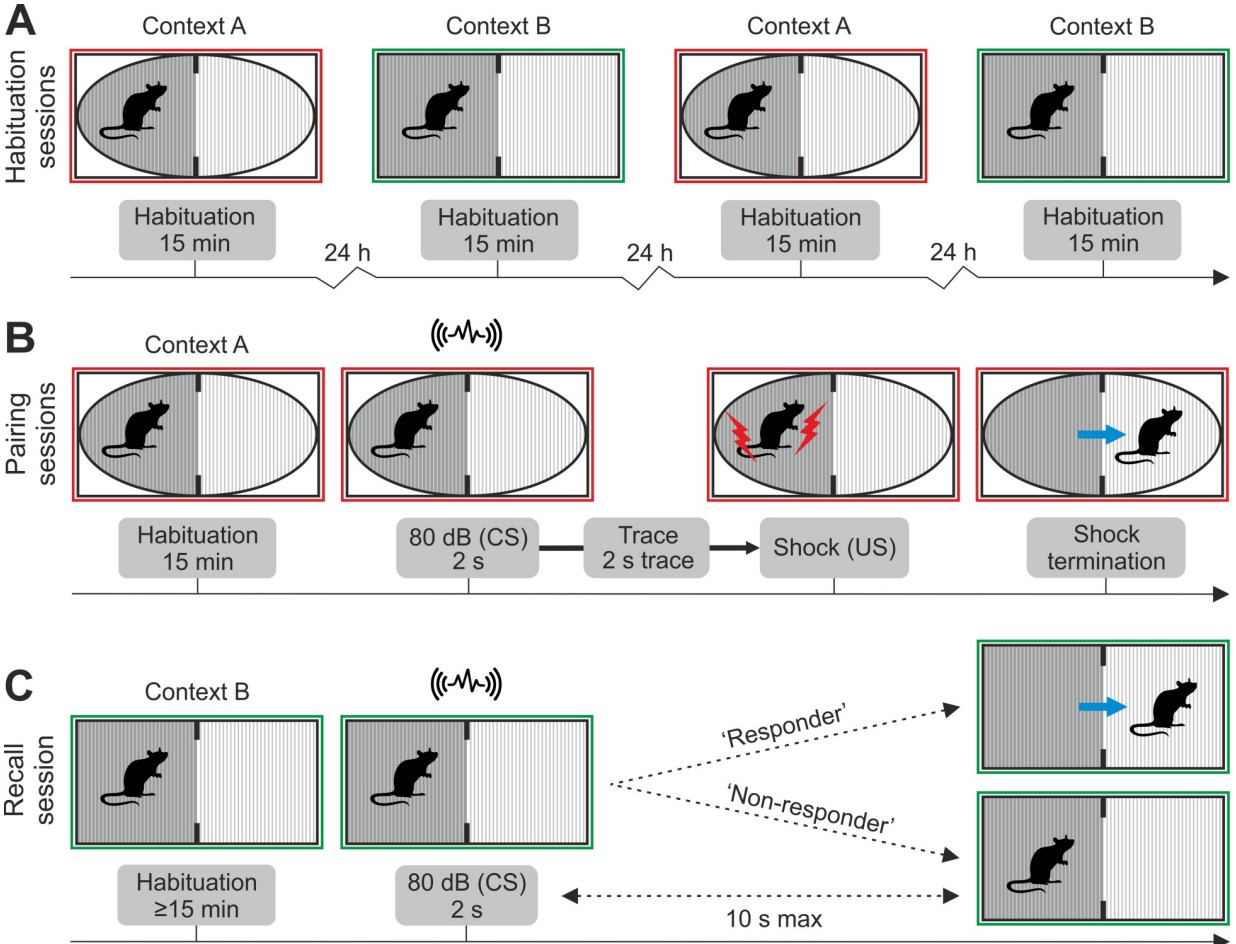

**Fig 1. Schematic overview of One-Trial Trace Escape Reaction (OTTER).** (**A**) A rat is initially habituated to environmental contexts A and B in alternating daily sessions. (**B**) During the pairing session, the rat hears a sound cue (CS) while in the dark chamber of one of the two contexts (context A or B); two seconds later, the rat receives a foot shock (US) that is terminated once the rat transfers to the light chamber. (**C**) The recall of CS-2s-US association is tested in the recall session, which occurs 24 hours later in the alternate context (context B or A). The CS is delivered when the rat settles in the dark chamber, and the rat's reaction is observed. Upon hearing the CS, the rat either escapes into the light chamber ('responder') or stays in the dark chamber ('non-responder').

## Methods

### Animals

Adult male Wistar rats (ENVIGO; 12–14 weeks old) were used in the experiment (n = 32). Upon arrival, the 28-day-old rats were housed in standard laboratory cages (50 x 25 x 25 cm), two animals per cage. Laboratory food and tap water were supplied *ad libitum*. The room where the animals were kept was ventilated with a constant temperature of 22˚C and 50% humidity. The rats were kept on a 12-hour light cycle, with light being turned on daily at 6 am.

Before the start of the OTTER task, all rats were handled by the experimenter for 3 minutes daily for four days. All experiments were conducted during the light phase of the day (9 am to 1 pm) because rats show lower locomotion during that time [19]. All animal procedures were approved by the Ethical Committee of the Czech Academy of Sciences and complied with the Animal Protection Act of the Czech Republic and EU directive 2010/63/EC.

## Apparatus

Two modified TSE multi-conditioning shuttle boxes (TSE Systems GmbH, Germany) were used in the experiment. Each shuttle box consisted of two interconnected 24 x 47 cm chambers. The first chamber of both shuttle boxes was built from transparent acrylic glass (light chamber), while the second chamber was created using dark opaque acrylic glass (dark chamber). A dark opaque lid was used to cover the dark chamber, resulting in light intensity of less than 3 lx in the chamber's center. The light chamber was left uncovered; moreover, we added another light source to reach a light intensity of 1090 lx in the chamber's center. Intense light is highly uncomfortable for rodents [20], which motivated the rats to spend most of their time in the dark chamber. The chambers were separated by a custom-made partition with a wide opening (4 x 40 cm central opening, custom-made) made of black acrylic glass.

The shuttle boxes were soundproofed and equipped with a speaker. Once triggered by the TSE software, the speaker delivered a 2400 Hz sound cue. The sound cue was delivered at 80 dB SPL intensity for 2 seconds. The walls of both chambers were equipped with infrared devices that registered the animal's location within the apparatus. The floor of both chambers consisted of a metallic grid with 0.5 cm diameter metal rods spaced 1.5 cm apart. When prompted, the metal rods delivered a 1.0 mA pulsatile electric stimulus with a 400 ms period (a 200 ms, 1 mA stimulus followed by 200 ms no stimulus) to the animal in the dark chamber.

The two TSE multi-conditioning shuttle boxes were made visually and olfactorily distinct so that one shuttle box served as environmental context A and the other as environmental context B. The walls of the light chamber were decorated with an aquarium scene on a circular insert in context A, while in context B, the walls of the light chamber were decorated with black stripes. Context A was cleaned with an alcohol-based wash, while context B was cleaned with a vinegar-based wash.

We presume the OTTER task can be conducted using any similar apparatus where the above-described general principles are adhered to and will deliver comparable results to those using TSE shuttle boxes.

## Habituation, pairing, and recall

Each rat was individually habituated to the environmental contexts A and B twice in a series of four 15-minute habituation sessions; only one habituation session took place every day, and the sessions in contexts A and B alternated daily. At the start of every habituation, each rat was placed in the dark chamber and was left to explore both chambers freely.

Following habituation, rats were conditioned to CS-2s-US pairing. The pairing took place in the same context as the first habituation session: rats first exposed to context A experienced CS-2s-US pairing in context A and vice versa for context B. The beginning of the pairing session closely resembled the habituation session, as rats were allowed to move freely through the apparatus for 15 minutes. At this point, rats did not transfer between the chambers at all or transferred only seldom. Following the 15-minute interval, a to-be-conditioned stimulus – a three-second sound cue – was delivered. We advise that the CS should be delivered with caution, as delivering the CS at an inappropriate moment might hamper the CS-2s-US acquisition. The rat must be located in the dark chamber, resting and not facing the opening in the partition between chambers (to avoid bias toward escaping through the 'door'). Two seconds after the CS stopped, an electric foot shock was delivered (US) to the rat by the metallic floor grid. The US was automatically terminated if the rat's position was registered in the light chamber or if the rat did not leave the dark chamber in 20 seconds. Rats that did not escape to the light chamber were excluded and did not proceed to the recall session. Rats that escaped

were returned to their home cage immediately after the escape and were left undisturbed for the next 24 hours, after which they were tested for recall.

The recall of the CS-2s-US pairing took place in the alternate context, i.e., if the pairing took place in context A, the recall was tested in context B. Recall session resembled the pairing session with the exception that the US was not delivered. The CS was delivered no sooner than after 15 minutes and only if the rat rested in the dark chamber. Following the CS delivery, the rat's response was observed. Rats that escaped to the light chamber within 10 seconds of the CS start were considered 'responders,' while those that remained in the dark chamber were considered 'non-responders.'

## Statistics

IBM SPSS Statistics (version 25.0, IBM Corp., 2017) was used to analyze behavioral data. More than half of the data did not meet parametric assumptions; therefore, we used non-parametric tests in all statistical analyses (Mann-Whitney test, Friedman test, Wilcoxon-signed rank test, Kruskal-Wallis test, Fisher's exact test). The significance threshold was set to p = 0.05.

## Data visualization

Data visualizations were created in IBM SPSS Statistics (version 25.0, IBM Corp., 2017), Corel, and R using the visualization library ggplot2 [21]. Heatmaps were obtained using the two-dimensional kernel density estimation function, kde2d, from the MASS library [22].

## Results

### Habituations

During the four 15-minute habituation sessions in environmental contexts A and B, rats preferred the dark chamber in both contexts (Fig 2A). We found that rats transferred significantly less often to the light chamber in context B than in context A during the first habituation session (Mann-Whitney test: hab 1: U = 55.00, $p$ = 0.008). The difference between the number of transfers to the light chamber in context A and B was not significant during any other habituation session (Fig 2B) (Mann-Whitney test: hab 2: U = 112.50, $p$ = 0.775; hab 3: U = 72.00, $p$ = 0.057; hab 4: U = 116.50, $p$ = 0.899). For plots of movement through the apparatus by the individual rats during the 15-minute habituation sessions, see S1 Fig. We found no significant difference in the time spent in the dark chamber (S2 Fig) (Mann-Whitney test: hab 1: U = 94.00, $p$ = 0.315; hab 2: U = 96.50, $p$ = 0.361; hab 3: U = 97.00, $p$ = 0.373; hab 4: U = 88.00, $p$ = 0.213) in context A and B during any of the four habituation sessions.

As each rat was habituated to the same context twice, we also assessed if the number of transfers to the light chamber and the time spent in the dark chamber differed between these two habituation sessions. We found that in group 1 (first habituation in context A) the number of transfers to the light chamber did not significantly change between the two sessions in either context A (Wilcoxon signed-rank test: Z = -0.473, $p$ = 0.658) or in context B (Z = -0.199, $p$ = 0.873). Similarly, we found no significant difference in the time spent in the dark chamber between the two sessions in neither context A (Wilcoxon signed-rank test: Z = -0.879, $p$ = 0.379) nor in context B (Z = -0.314, $p$ = 0.753) in group 1. In group 2 (first habituation in context B), we observed similar behavior: the number of transfers to the light chamber did not significantly change between the two habituations sessions in context A (Wilcoxon signed-rank test: Z = -0.063, $p$ = 0.964) nor in context B (Z = -1.129, $p$ = 0.283). Time spent in the dark chamber also did not significantly differ between the two sessions in the same context

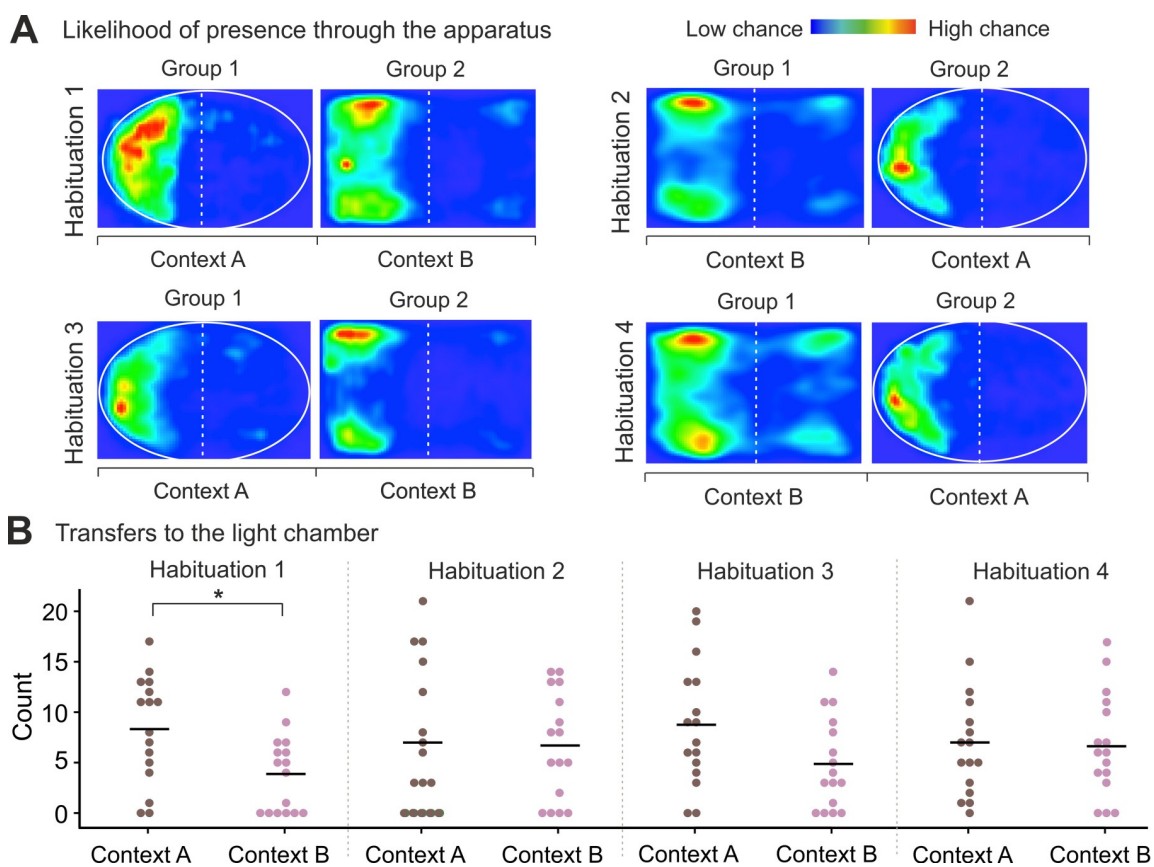

**Fig 2. Chamber preference and transfers to the light chamber during 15 min habituation sessions.** (**A**) Heat maps showing the probability of occurrence of the animals during each habituation session; blue indicates minimal presence, and red signifies a frequent stay. Rats were most often present in the dark chamber of contexts A and B during each session; the time spent in the dark chamber did not significantly differ between contexts A and B during habituation sessions. (**B**) Transfers to the light chamber in contexts A and B during habituation sessions. Rats transferred to the light chamber significantly less in context B than in context A during the first habituation session (p < 0.01), but not during any other habituation session.

(Wilcoxon signed-rank test: Z = -0.549, *p* = 0.583; Z = -0.879, *p* = 0.379 for context A and context B, respectively).

## Pairing sessions

At the beginning of the pairing session, 24 rats were randomly selected to be presented with CS-2s-US (test group) and 8 rats to be presented with CS only (control group). One rat in the test group was excluded at the beginning of the pairing session as it lingered in the light chamber, and CS-2s-US could not be presented. All other test group rats (N = 23) were presented with CS-2s-US and escaped to the light compartment within 20 seconds. The average latency to escape was 10.0 ± 0.9 seconds (SEM) from the CS onset. Rats that responded to CS during the recall the next day escaped the US on average slightly faster than future non-responders (9.3 s compared to 11.0 s) (Fig 3C); however, this difference was not statistically significant (Mann-Whitney test: U = 29.000, *p* = 0.601). None of the control rats (N = 8) transferred to the light chamber upon hearing the CS.

During the habituation session preceding recall, rats transferred to the light chamber significantly less than during habituation before CS-2s-US pairing (Wilcoxon signed-rank test: Z = -2.762, *p* = 0.004) (pairing session habituation Mdn = 5.5, recall session habituation Mdn = 3)

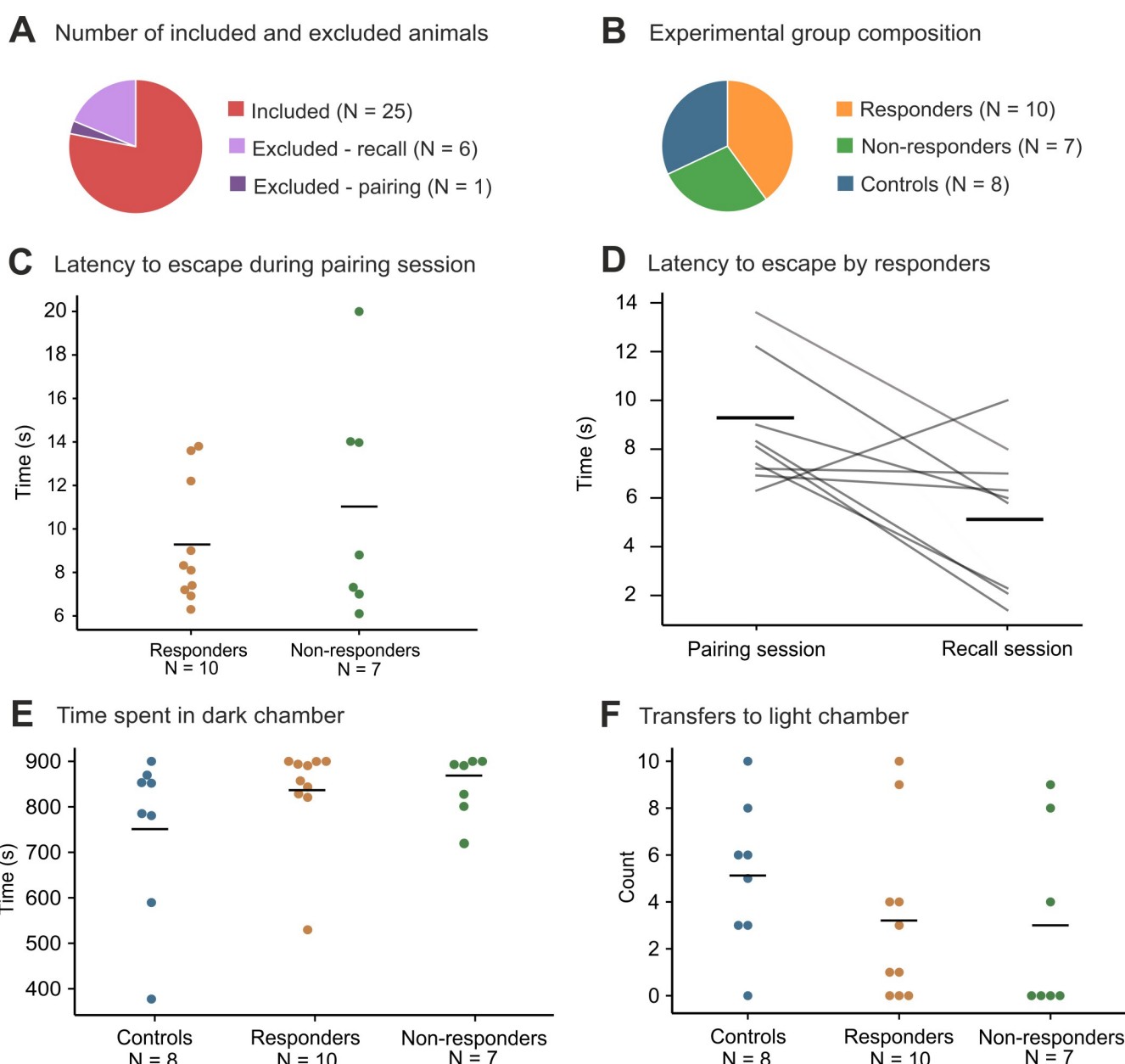

**Fig 3. The OTTER task normative data.** (**A**) The pie chart illustrates animals included in the study (N = 25) and animals excluded before the pairing (N = 1) and the recall session (N = 6). (**B**) Ratios of responders (N = 10), non-responders (N = 7), and controls (N = 8) in the group of 25 animals included in the study. Animals in the control group did not respond to CS during the recall phase. (**C**) Latencies to transfer to the light chamber during the pairing (since CS-2s-US start) by responders and non-responders. Non-responders transferred to the light chamber after 11.0 ± 1.9 seconds and responders after 9.3 ± 0.9 seconds. (**D**) Latencies to transfer to the light chamber by responders during pairing (since CS-2s-US start) and recall (since CS start; CS length was 2 seconds). Responders transferred to the light chamber after 9.3 ± 0.9 seconds during the pairing and after 5.1 ± 0.9 seconds during the recall. (**E**) Time spent in the dark chamber during the habituation preceding recall by non-responders, responders, and controls. (**F**) Transfers to the light chamber during the habituation preceding recall by non-responders, responders, and controls.

(Fig 4A). Although rats appeared to spend more time in the dark chamber during habituation preceding recall (Mdn = 843.8 s) than during habituation before pairing (Mdn = 814.4 s), the difference was not significant (Wilcoxon signed-rank test: Z = -0.843, $p$ = 0.410) (Fig 4B).

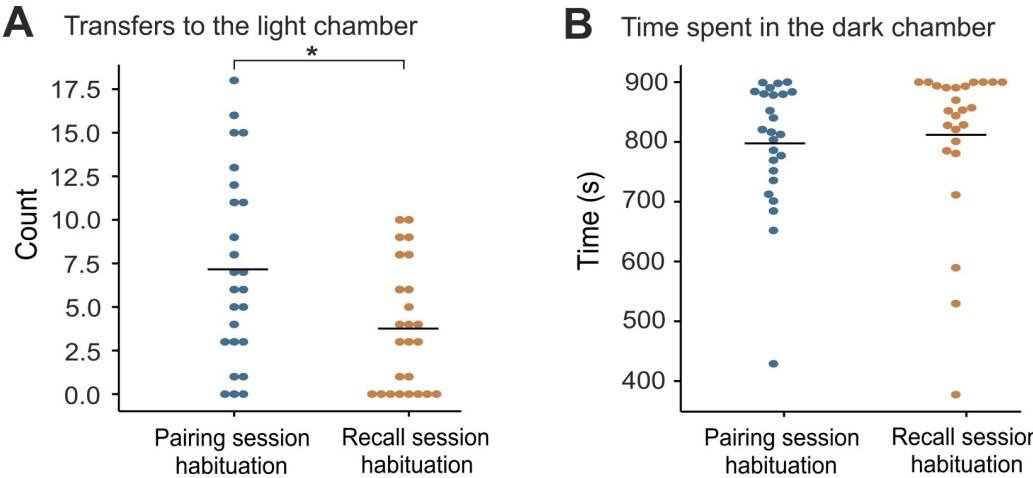

**Fig 4. Comparison of transfers to the light chamber and time spent in the dark chamber during pairing and recall session.** (**A**) Transfers to the light chamber during pairing and recall session habituations (data from contexts A and B combined). Rats transferred to the light chamber significantly less during recall session habituation (p < 0.01). (**B**) Time spent in the dark chamber during pairing and recall session habituations (data from contexts combined). The difference in time spent in the dark chamber between these two sessions did not significantly differ.

### Recall sessions

During the recall session, the CS was successfully presented to 17 rats from the test group and 8 animals from the control group; 6 rats from the test group were excluded from the experiment as they lingered in the light chamber during the time of the intended CS presentation (Fig 4A). In the test group, 59% (N = 10) of rats moved to the light chamber within 10 seconds of the start of CS (Fig 4B). The average time to escape was 5.1 ± 0.9 seconds (SEM) from the CS onset (Fig 4D). None of the control rats (N = 8) transferred to the light chamber in response to the CS. The number of responders in the test group was significantly higher than in the control group (Fisher's exact test: p (two-tailed) = 0.008). We found no statistically significant difference in the time spent in the dark chamber (Kruskal-Wallis test: H(2) = 1.674, p = 0.433) (Fig 4E) or in the number of transfers to the light chamber (Kruskal-Wallis test: H(2) = 1.384, p = 0.501) between responders, non-responders and controls during the habituation session before recall (Fig 4F).

### Discussion

We designed a behavioral task–OTTER–in which a single event is sufficient for an animal to incidentally associate two novel events separated by a time gap. The animal later actively demonstrates the association in a way that allows unambiguous evaluation of memory acquisition while retaining balance between successful performers vs. non-performers. The OTTER task is a trace-conditioning task that employs active avoidance to capture an important feature of naturalistic human episodic memory: the incidental memory acquisition of an event experienced only once. OTTER utilizes species-specific preferences and naturally occurring behavior, which greatly reduces the amount of training needed and increases the task's ecological validity.

Here we put the OTTER task in the context of other rodent behavioral paradigms that utilize active avoidance, one-trial learning, and trace conditioning: the elements from which the OTTER task is built. We also discuss the advantages that the OTTER task offers as a paradigm

for studying elementary features or prerequisites of human episodic memory in rodents as well as its limitations.

One of the characteristics of the OTTER task is active avoidance. Active avoidance tasks provide a discrete measure of memory acquisition (for an in-depth review, see [23,24]). The most commonly used paradigm in the study of active avoidance is the shuttle-box avoidance task. The apparatus most often used in this task consists, similar to the OTTER task, of two chambers between which the animal can freely transfer. When a warning signal (CS) is presented, the animal needs to transfer to the other chamber during a scheduled time interval to terminate or avoid a foot shock (US) – an aspect that the OTTER task shares as well. One of the major differences is that in the shuttle-box avoidance task, CS can be presented regardless of the animal's location within the apparatus, while in the OTTER task, CS is presented only when the animal resides in the dark chamber. Another difference is that in the shuttle-box avoidance task, animals learn to avoid the foot shock by transferring to the other chamber during the scheduled interval only after numerous CS-US pairings: multiple CS-US pairings are delivered per session and the training usually takes 7–10 days; in contrast, rats learn to associate CS with the US in the OTTER task after only a single pairing. A substantial advantage that active avoidance tasks share with the OTTER task is the discrete outcome, which we discuss in a later section. Diehl et al. [25] highlight three limitations of the shuttle-box avoidance paradigm: (i) lack of safe location hinders clear behavioral output, low ecological validity due to (ii) no-cost of avoidance and due to (iii) avoidance behavior terminating the threat. Here, we discuss each limitation of the shuttle-box paradigm and assess the OTTER on each of the three points. The first limitation is that there is no permanently safe location in the shuttle-box paradigm hence animals often display freezing behavior, which is a natural response of rats to anxietyy or unavoidable threats. Freezing is most common at the beginning of training and is later replaced by an escape reaction in ~75% of rats. Freezing therefore interferes with active avoidance either temporarily or, in some animals, permanently. In comparison, the OTTER task provides a permanently safe, although unpreferred, location – the brightly lit chamber that is always available to the animal. The second limitation of the shuttle-box paradigm is that no-cost is associated with avoidance behavior, which decreases the ecological validity of the task. In their natural habitat, animals engage in anti-predatory avoidance only when necessary or if the risk is high because avoidance precludes other important behaviors such as foraging or mating [26]. In comparison, unnecessary avoidance is rare during the test phase in the OTTER task because it is associated with the cost of exchanging naturally preferred dark for a brightly lit environment. The third limitation of the traditional shuttle-box paradigm marked by Diehl et al. is that avoidance terminates the warning signal (CS). Therefore, exposure to the CS differs among individuals. In the OTTER task, the animal has no control over the termination of the CS: the CS terminates after 2 seconds regardless of the rat's behavior, giving all animals the same exposure to the CS. Overall, the OTTER task performs well on each point mentioned by Diehl et al. [25]; the last two points contribute to the ecological validity of the OTTER task.

Tasks of one-trial memory acquisition for rodents that do not require extensive training include conditioned taste aversion, novel-object recognition, and contextual fear conditioning. Despite their undoubtful usefulness and productive usage in neuroscience, these tasks have some limitations when considering incidental memory that could be related to episodic memory. The behavior in these tasks does not reflect the all-or-none threshold retrieval dynamics (presence or absence of retrieval–a discrete measure of outcome) observed in human memory [27]. The conditioned taste aversion is an excellent task with a long delay between the CS (taste) and US (poisoning). However, since animals can also be unconscious during the US and still form an association [28,29], the conditioned taste aversion might be considered an

implicit memory where no recollection is needed for its expression. Novel object recognition tasks [30–32] were developed specifically to test episodic-like memory; however, the tasks received criticism that they rely more on a sense of familiarity rather than on recollection [33]. In addition, the outcome measures in novel object recognition tasks (time spent exploring the object) and contextual fear conditioning tasks (freezing) are continuous, making it difficult to reliably single out 'learners' from 'non-learners'. This allows for studying neurophysiological changes in both groups,.

Trace fear conditioning tasks can be used as tests for basic processes essential for forming episodic-like memory due to their temporal structure, which involves a time gap between the CS and US. However, the CS-trace-US contingency is presented more than once [34,35]. Although rodents may associate CS with the US in trace fear conditioning after a single CS-trace-US exposure, there is no observable variable that could indicate the acquired knowledge. Specifically, the outcome measured in trace fear conditioning and similar tasks, freezing, is a response associated with an absence of an escape route [36] that might indicate behavioral despair as no action can avert the stressor. In this regard, freezing might, at first, be a non-specific defensive response that does not reflect the acquisition of knowledge about the contingency. It is also possible that freezing behavior emerges only after repeated CS-trace-US presentation. From this perspective, the OTTER task bypasses the potentially non-specific freezing response because a single pairing session results in a clear avoidance response during the recall session. In addition, the brain regions involved in avoidance behavior change with repeated exposure to the aversive stimulus throughout training [37]; for this reason, a behavioral task that establishes memory incidentally after a single exposure might better reflect neural activity that is involved in the acquisition of episodic memory.

One of the practical aspects of the OTTER tasks is that it does not need pre-training or 'priming' the animals to anticipate a behaviorally relevant event. The pre-training in existing rodent paradigms is ~~can be~~ twofold: shaping the desired response behavior or repeated exposure to stimuli. Both types of pre-training create an expectancy of contingency. For example, in trace fear conditioning, the CS-trace-US is presented repeatedly. It could be argued that repeated habituation in the OTTER taks is also 'pre-training'. However, repeated habituation to the environment without either US or CS is not pre-training that creates an expectancy for contingency; habituation simply reduces the incidence of variable behaviors.

Another advantage of the OTTER task is that active avoidance behavior offers an unambiguous binary measure of recall so that each animal can be confidently singled out as a 'responder' or 'non-responder.' In contrast, fear conditioning tests and novel object recognition tasks use a continuous variable as an indicator of learning, such as freezing or duration of exploration. When a continuous variable is used to classify animals into discrete groups, a threshold value is needed. The threshold is usually more or less arbitrary, and even then, it is often unclear how to classify animals just around the threshold values. Using a continuous outcome variable as a basis for classification might therefore provide unclear results. Rodent tasks with unambiguous output variables usually involve showing the knowledge by entering a correct place or pressing a correct button [6,38]. However, extensive pre-training is often required since the response must be shaped first (usually approach behavior). The need for long pre-training to achieve the response precludes the study of incidentally acquired memories. In the OTTER task, rats naturally exhibit the chosen declarative behavior without prior training.

As long as the general principle of the OTTER task is adhered to, that is, controlling animal behavior by balancing conflicting species-specific behavioral tendencies, the OTTER task is flexible and can be embodied even by different physical instances. We are currently developing a second variant of the OTTER task with the working title 'cold-OTTER.' In the 'cold-OTTER' task, the invariant behavior is achieved by utilizing the rat's and mice's preference for warmth,

which results in the avoidance of the cold sub-area of the apparatus. The flexible nature of the OTTER task allows for adapting the task for different species and research contexts.

The OTTER task offers a high temporal precision of the recall event, making it an auspicious task for detailed studies of retrieval mechanisms. There is only a brief time window when an animal retrieves information and acts upon it. Such pinpointing of the recall event is difficult in tasks where a behavioral response is registered as a frequency of behavior during a time interval (freezing or exploration duration). The temporal precision of the recall event offered by the OTTER task can be especially advantageous if combined with high temporal resolution methods, such as electrophysiology [39] or calcium imaging [40]. The OTTER task might therefore serve as a valuable behavioral paradigm for a detailed study of neural mechanisms involved in incidental learning and trace conditioning.

Although we do not consider OTTER to be an episodic-like memory task according to the present criteria, the underlying mechanisms may be related to some essential elements of episodic memory. The successful recall of CS-2s-US in OTTER seems to meet several criteria of such elements: a) the memory was incidentally encoded [7], b) encoding occurred after a single exposure [41], c) there was no pre-training involved [42], d) rat behavior observed threshold retrieval dynamics [27], and e) rats were able to retrieve the information flexibly in a different context [43]. However, the OTTER task does not meet the episodic-like memory criterion of demonstration of what-where-when knowledge of past experiences [44] because the flight in response to the CS cannot indicate if the rat remembers where and when it experienced CS-2s-US.

The OTTER task could be utilized to study the extinction of incidentally acquired memory based on a single exposure. This aspect is highly relevant to several neuropsychiatric disorders, especially post-traumatic stress disorder (PTSD). In this sense, the OTTER task could serve as an ecologically valid memory acquisition/extinction model of PTSD. We expect the extinction curve of CS-2s-US association could be influenced in both directions (faster/slower) by behavioral manipulations during or after the recall session.

As in any behavioral task, there are limitations to the OTTER task. First, the one-trial nature of the OTTER task precludes repeated measurements often required to accumulate sufficient amounts of data (e.g., in electrophysiology). This limitation stems from probing the incidental one-trial aspects of episodic-like memory and seems unavoidable. Second, it cannot be ruled out that 'non-responders' did form the CS-2s-US association but failed to act upon it, or did not generalize the experience beyond the original context. In assessing the recall, we rely on motoric output that is only indirectly related to the animal's mental state. However, we found no significant difference in time spent freezing at the start of the recall session (S3 Fig) and following the CS presentation (S3 Fig) between 'non-responders,' 'responders,' and controls in our preliminary experiments. Our results suggest that 'non-responders' did not associate the US with either CS or the dark chamber.

In conclusion, we designed a trace conditioning task called OTTER that is adaptable, gives rapid results, and is easy to conduct. Due to the association of temporarily discontinuous events, the OTTER task can improve our understanding of the neural mechanisms of trace conditioning and possibly memory extinction. The behavioral response in the OTTER task is ecologically valid because it takes advantage of the natural behavioral tendencies of rodents. The OTTER task is also ecologically valid in relation to human incidental memory: recall in incidental memory tasks was reported to be 35%-90%, depending on the task [45–47]. We demonstrated that rats could utilize the knowledge acquired from a single experience and use it to their advantage in a different context: rats demonstrated the same behavior that resulted in the termination of the unpleasant stimulus they experienced 24 hours earlier. We observed no 'ceiling effect' and a good balance between 'responders' and 'non-responders' (close to 1:1),

which may be valuable during the tracing of neural changes in the early ~~episodic-like~~ memory acquisition. The OTTER task extends the current range of trace conditioning tasks, capturing the one-trial and incidental nature of encoding, and offers high temporal precision regarding when the memory recall occurred. The OTTER task shares aspects with episodic memory due to its incidental, single-trial character with minimal training requirements.

## Supporting information

**S1 Fig. Movement of each rat through the apparatus during four 15-minute habituation sessions.** Each horizontal line corresponds to one animal. Black represents a stay in the dark chamber; gray represents a stay in the light chamber. Symbols "A" and "B" on the vertical axis correspond to the environmental context of the rat's habituation. Context A was oval-shaped and cleaned with an alcohol-based wash, while context B was rectangular-shaped and cleaned with a vinegar-based wash.
(TIF)

**S2 Fig. Time spent in chambers of contexts A and B by rats during habituation sessions.** Each rat (N = 32) received two habituation sessions in each context in an alternating manner. Starting context was chosen randomly for each rat. Rats preferred the dark chamber and spent very little time in the light chamber in both contexts across habituation sessions. The difference in the time spent in the dark chamber was significantly higher than the time spent in the light chamber in each context and during all five habituation sessions (Mann-Whitney test: Hab 1 context A: U = 0.000, $p$ = 0.000; Hab 1 context B: U = 0.000, $p$ = 0.000; Hab 2 context A: U = 1.000, $p$ = 0.000, Hab 2 context B: U = 0.000, $p$ = 0.000; Hab 3 context A: U = 1.000, $p$ = 0.000, Hab 3 context B: U = 6.000, $p$ = 0.000; Hab 4 context A: U = 1.000, $p$ = 0.000, Hab 4 context B: U = 0.000, $p$ = 0.000. Error bars indicate SEM, and * indicates $p < 0.05$.
(TIF)

**S3 Fig. Freezing at the beginning of the recall session and after the CS presentation.** Freezing was assessed manually from video recordings by a blinded experimenter. As a freezing, we considered any lack of movement except for breathing. **(A)** Time spent freezing (s) during the first minute of the recall session. There was no significant difference in time spent freezing between 'responders,' 'non-responders,' and control rats (Kruskal-Wallis test: H = 6.436, p = 0.092). **(B)** Time spent freezing (s) during one minute after CS presentation. We found no significant difference in time spent freezing between 'non-responders' and control rats (Mann-Whitney test: U = 11.00, p = 0.831). This suggests that 'non-responders' did not recall the CS-2s-US association.
(TIF)

## Acknowledgments

We would like to thank David Levcik for valuable feedback on the initial draft of the manuscript.

## Author Contributions

**Conceptualization:** Dominika Radostova, Branislav Krajcovic, Hana Brozka.

**Funding acquisition:** Ales Stuchlik.

**Investigation:** Dominika Radostova, Daniela Kuncicka.

**Methodology:** Dominika Radostova, Daniela Kuncicka.

**Resources:** Jan Svoboda, Ales Stuchlik.

**Supervision:** Ales Stuchlik, Hana Brozka.

**Validation:** Lukas Hejtmanek.

**Visualization:** Daniela Kuncicka, Hana Brozka.

**Writing – original draft:** Dominika Radostova, Branislav Krajcovic, Hana Brozka.

**Writing – review & editing:** Daniela Kuncicka, Branislav Krajcovic, Lukas Hejtmanek, Tomas Petrasek, Hana Brozka.

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
