## [Decision Letter · Decision Letter 0]

31 Oct 2022

PONE-D-22-24003Incidental Temporal Binding in Rats: A Novel Behavioral Task Relevant to Episodic MemoryPLOS ONE

Dear Dr. Brozka,

Thank you for submitting your manuscript to PLOS ONE. After careful consideration, we feel that it has merit but does not fully meet PLOS ONE’s publication criteria as it currently stands. Therefore, we invite you to submit a revised version of the manuscript that addresses the points raised during the review process. We ask you in particular to address the four points raised by Reviewer #1.

We look forward to receiving your revised manuscript.

Kind regards,

Robert Sutherland, Ph.D

Academic Editor

PLOS ONE

Journal Requirements:

"This work was supported by Czech Science Foundation (GACR) grant 20-00939S awarded to A.S. www.gacr.cz

Institutional support for IPHYS was provided by RVO: 67985823."

"This work was supported by Czech Science Foundation (GACR) grant 20-00939S. Institutional support for IPHYS was provided by RVO: 67985823."

"This work was supported by Czech Science Foundation (GACR) grant 20-00939S awarded to A.S. www.gacr.cz

Institutional support for IPHYS was provided by RVO: 67985823."

Reviewers' comments:

Reviewer's Responses to Questions

**Comments to the Author**

1. Is the manuscript technically sound, and do the data support the conclusions?

Reviewer #1: Partly

Reviewer #2: No

2. Has the statistical analysis been performed appropriately and rigorously? 

Reviewer #1: Yes

Reviewer #2: Yes

3. Have the authors made all data underlying the findings in their manuscript fully available?

Reviewer #1: Yes

Reviewer #2: Yes

4. Is the manuscript presented in an intelligible fashion and written in standard English?

Reviewer #1: Yes

Reviewer #2: Yes

5. Review Comments to the Author

Reviewer #1: The experiments reported in this manuscript were directed at rodent task development in the learning and memory domain. Specifically, the authors were interested in developing a behavioural task to assess episodic memory in the rat with an emphasis on a task in which the experience occurs only once, an important component of episodic memory.

In my opinion, the work is well-motivated, and the authors have developed an interesting task with some strong design features. The results are interesting and clearly presented and analyzed.

I do have a couple of significant issues with the experiments and manuscript.

First, I think it is a bit of a stretch to suggest that this task is tapping into the temporal component of episodic memory in which the organism must learn about when the event occurred. This paradigm simply utilizes a standard Pavlovian conditioning procedure with a very short trace (2 seconds). I don’t see any evidence presented that they learned this temporal information, just that they learned the association between the CS and US.

Unless I am missing something here this would change some of the content of the introduction and discussion.

Second, the fact that only 59% of the subjects learned the task is a positive and a negative. It is the former for a variety of reasons including the ability to understand individual differences in one-trial learning. It is a negative, in my view because it is probably the case if the whole group of subjects conditioning results was graphed and analyzed together there would be no effect. One potentially helpful approach would be to measure heart rate conditioning simultaneously. What you might find is that most of the subjects did learn the CS-US association but not all of them showed it in their avoidance behaviour. I understand the freezing behaviour presented suggests otherwise but conditioning measured using freezing might require multiple trials and/or several reinforcers to emerge.

Third, one of the main issues the authors raise about this field of enquiry is that many episodic paradigms using rodents requires shaping, training, and reinforcement, etc. The authors claim that their paradigm avoids these issues. Clearly, this is true concerning some of these design features, but I would consider the repeated pre-exposure phase as training that is important to show this one-trial learning.

Finally, I think a stronger demonstration of scholarship is needed. There have been a variety of tasks designed to assay episodic memory in rodents, birds, and monkeys and this work should be integrated.

Reviewer #2: The authors describe a one-trial avoidance learning task that incorporates a brief trace (2 sec) between the presentation conditioned stimulus and unconditioned stimulus. The authors argue that because of this trace, the single learning trial, and that the memory test is performed in a different context that the memory is episodic.

The behavioural task is somewhat interesting, but the argument that it demonstrates episodic memory is weak and not compelling. Rather than demonstrating that the memory meets the standard features of episodic memory (what, where, and when), they limit their argument to what they term incidental learning. However, their task does not offer much novelty or more than other tasks:

one-trial inhibitory avoidance (Parent, M. B., Avila, E., & McGaugh, J. L. (1995). Footshock facilitates the expression of aversively motivated memory in rats given post-training amygdala basolateral complex lesions. Brain Res, 676(2), 235-244.)

One-trial socially transmitted food preference (Winocur, G., McDonald, R. M., & Moscovitch, M. (2001). Anterograde and retrograde amnesia in rats with large hippocampal lesions. Hippocampus, 11(1), 18-26.)

The authors must have a proof of concept demonstrating that the task involves episodic memory. They should also demonstrates that lesions that impair episodic memory in humans also impair memory in this task.

The authors should report on longer trace intervals than 2 seconds. How does performance relate to the duration of the trace.

The task is not very pragmatic as only 50% of the rats show the desired learning and avoidance behaviour.

In sum, this is a methods paper and insufficient for publication. It would be best if accompanied with an experimental manipulation (e.g., lesion or inactivation).

6. PLOS authors have the option to publish the peer review history of their article (what does this mean?). If published, this will include your full peer review and any attached files.

Reviewer #1: **Yes: **Dr. Robert McDonald

Reviewer #2: No

---

## [Author Response · Author response to Decision Letter 0]

2 Dec 2022

Reviewer #1: The experiments reported in this manuscript were directed at rodent task development in the learning and memory domain. Specifically, the authors were interested in developing a behavioural task to assess episodic memory in the rat with an emphasis on a task in which the experience occurs only once, an important component of episodic memory.

In my opinion, the work is well-motivated, and the authors have developed an interesting task with some strong design features. The results are interesting and clearly presented and analyzed.

I do have a couple of significant issues with the experiments and manuscript.

First, I think it is a bit of a stretch to suggest that this task is tapping into the temporal component of episodic memory in which the organism must learn about when the event occurred. This paradigm simply utilizes a standard Pavlovian conditioning procedure with a very short trace (2 seconds). I don’t see any evidence presented that they learned this temporal information, just that they learned the association between the CS and US.

Unless I am missing something here this would change some of the content of the introduction and discussion.

Radostova et al., :Thank you for the positive feedback and the comment. Indeed, we cannot judge if the organism ‘learns when the event occurred'. We changed parts of the introduction and discussion and hope they are not misleading anymore. Changes related to this issue are highlighted in red in the attached manuscript file. We added a sentence that temporal binding is assessed using trace-conditioning tasks (in red; lines 31-32, line 415). We changed references to ‘knowledge acquired” to ‘formation’ of the association (lines 245-246, line 356) We hope we responded to this question appropriately.

Reviewer #1: Second, the fact that only 59% of the subjects learned the task is a positive and a negative. It is the former for a variety of reasons including the ability to understand individual differences in one-trial learning. It is a negative, in my view because it is probably the case if the whole group of subjects conditioning results was graphed and analyzed together there would be no effect. One potentially helpful approach would be to measure heart rate conditioning simultaneously. What you might find is that most of the subjects did learn the CS-US association but not all of them showed it in their avoidance behaviour. I understand the freezing behaviour presented suggests otherwise but conditioning measured using freezing might require multiple trials and/or several reinforcers to emerge.

Radostova et al., :Thank you for the comment and, yes, we agree. We tried measuring the heart rate in the past during the OTTER learning and recall but a signal from our wireless telemetry probes was canceled out by experiments being conducted in an electrified box. In the manuscript, we erased the part where we claim that responders and non-responders can be confidently classified (lines 425-427). 

Moreover (this overlaps with the response to the other reviewer’s comment), the percentage of responders in the OTTER corresponds to the number of humans that successfully recall the memory following incidental learning in an experimental setting (in humans the successful recall occurred little more often, in about 70% of cases). For this reason, we believe that (1) there is a likelihood that there will be rats without the memory of the CS-2s-US in the OTTER and (2) that the OTTER is an ecologically valid task. We added the information about human tasks to the text (in purple; lines 417-418):

“It is also ecologically valid in relation to human incidental memory: recall in incidental memory tasks was reported to be 35%-90%, depending on the task [41–43].”

Again, thank you for the comment, and, yes, it would be great if we could have heart rate data.

Reviewer#1: Third, one of the main issues the authors raise about this field of inquiry is that many episodic paradigms using rodents require shaping, training, reinforcement, etc. The authors claim that their paradigm avoids these issues. Clearly, this is true concerning some of these design features, but I would consider the repeated pre-exposure phase as training that is important to show this one-trial learning.

Radostova et al., : Thank you for the comment. We did not consider habituation to the apparatus to be pretraining. However, your point is a very good one and we added your suggestion to the text and explained why we do not agree (in blue; lines 331-333):

“It could be argued that repeated habituation is also 'pre-training'. However, repeated habituation is not pre-training in the sense that it creates the expectancy for the contingency; habituation simply reduces the incidence of variable behaviors"

But, we also narrowed the term pre-training in the introduction (in blue; line 36):

“We believe that the acquisition of episodic-like memory should not require conditioning or pre-training to to-be-learned information; moreover, evidence suggests that mechanisms of incidental memory acquisition might differ from acquisition with intent”

Finally, I think a stronger demonstration of scholarship is needed. There have been a variety of tasks designed to assay episodic memory in rodents, birds, and monkeys and this work should be integrated.

Thank you for the suggestion. We tried our best and added a section to the discussion of the manuscript. In short, we discuss how OTTER is related to avoidance learning and other behavioral tasks. As the section is lengthy, please see the manuscript (in green; lines 253-319). Hopefully, you will find the section to your liking.

Reviewer #2: The authors describe a one-trial avoidance learning task that incorporates a brief trace (2 sec) between the presentation conditioned stimulus and unconditioned stimulus. The authors argue that because of this trace, the single learning trial, and that the memory test is performed in a different context that the memory is episodic.

The behavioural task is somewhat interesting, but the argument that it demonstrates episodic memory is weak and not compelling. Rather than demonstrating that the memory meets the standard features of episodic memory (what, where, and when), they limit their argument to what they term incidental learning. However, their task does not offer much novelty or more than other tasks:

one-trial inhibitory avoidance (Parent, M. B., Avila, E., & McGaugh, J. L. (1995). Footshock facilitates the expression of aversively motivated memory in rats given post-training amygdala basolateral complex lesions. Brain Res, 676(2), 235-244.)

One-trial socially transmitted food preference (Winocur, G., McDonald, R. M., & Moscovitch, M. (2001). Anterograde and retrograde amnesia in rats with large hippocampal lesions. Hippocampus, 11(1), 18-26.)

The authors must have a proof of concept demonstrating that the task involves episodic memory. They should also demonstrate that lesions that impair episodic memory in humans also impair memory in this task.

Radostova et al.,: Thank you for the comment. During the writing of the manuscript, we were cautious not to over-interpret the OTTER task as an episodic-like memory task. For that reason we used phrases such as ‘the OTTER is relevant to episodic memory’ and that it ‘models several aspects of the episodic memory’ (samples highlighted in yellow). A behavioral task does not have to model all aspects of episodic memory to be useful in learning about episodic memory. This is why ‘episodic memory’ is often mentioned in the text. However, we scrutinized the text and found some statements that could be interpreted as presenting the OTTER to be an episodic memory task. We added an explicit statement that we do not consider the OTTER to test true episodic memory (lines 374-375):

“Although we do not consider OTTER to be a true episodic-like memory task according to present criteria, we consider OTTER to be highly relevant to episodic memory”

Thank you for pointing out these issues.

The two other tasks that you mentioned are similar in several aspects but different in others. Parent et al., 1995 harness light and dark chambers in a similar manner as the OTTER, but it does not include the trace-conditioning aspect. We did not know this article before and we are happy that we are on a right track. Winocour et al., 2001 involve one-trial incidental learning, albeit social, but this task also does not involve the association of two events. Moreover, the behavior is likely evolutionary wired. The task is very interesting.

Reviewer #2: The authors should report on longer trace intervals than 2 seconds. How does performance relate to the duration of the trace.

Radostova et al.,: Thank you for the suggestion. We did not test longer intervals, as we wanted to have the highest yield of the learning rats possible. We presume that the performance will deteriorate with the increase of the trace interval, as CS-2s-US is presented only once in the OTTER task. Moreover, it is possible that with the increasing length of trace the learning shifts towards dissociation between CS and US (doi: 10.1016/j.beproc.2013.08.016), which we were not aiming for.

Reviewer #2: The task is not very pragmatic as only 50% of the rats show the desired learning and avoidance behaviour.

Radostova et al.,: That is true. However, 50% is sufficient if differences in neurobiology between responders and non-responders are the aim of the research. From our perspective, the non-responders are the best control group for the responders. Moreover, this level of performance is ecologically valid in relation to human episodic memory. Based on your comment we added a line in a discussion (highlighted in purple, lines 417-418):

“It is also ecologically valid in relation to human incidental memory: recall in incidental memory tasks was reported to be 35%-90%, depending on the task [41–43].”

Reviewer #2: In sum, this is a methods paper and insufficient for publication. It would be best if accompanied with an experimental manipulation (e.g., lesion or inactivation).

Radostova et al.,: We appreciate your opinion.

---

## [Decision Letter · Decision Letter 1]

2 Feb 2023

PONE-D-22-24003R1Incidental Temporal Binding in Rats: A Novel Behavioral Task Relevant to Episodic MemoryPLOS ONE

Dear Dr. Brozka,

Thank you for submitting your manuscript to PLOS ONE. After careful consideration, we feel that it has merit but does not fully meet PLOS ONE’s publication criteria as it currently stands. Therefore, we invite you to submit a revised version of the manuscript that addresses the points raised during the review process.

 Both of the reviewers find that you have not adequately situated your findings or discussion in the context of episodic memory and/or Pavlovian conditioning paradigms. If you are able to address their concern about this issue, please submit a revised manuscript.

Please submit your revised manuscript within six months.. If you will need more time than this to complete your revisions, please reply to this message or contact the journal office at plosone@plos.org. Please include the following items when submitting your revised manuscript:A rebuttal letter that responds to each point raised by the academic editor and reviewer(s). You should upload this letter as a separate file labeled 'Response to Reviewers'.A marked-up copy of your manuscript that highlights changes made to the original version. You should upload this as a separate file labeled 'Revised Manuscript with Track Changes'.An unmarked version of your revised paper without tracked changes. You should upload this as a separate file labeled 'Manuscript'.

We look forward to receiving your revised manuscript.

Kind regards,

Robert Sutherland, Ph.D

Academic Editor

PLOS ONE

Reviewers' comments:

Reviewer's Responses to Questions

**Comments to the Author**

1. If the authors have adequately addressed your comments raised in a previous round of review and you feel that this manuscript is now acceptable for publication, you may indicate that here to bypass the “Comments to the Author” section, enter your conflict of interest statement in the “Confidential to Editor” section, and submit your "Accept" recommendation.

Reviewer #1: (No Response)

Reviewer #2: (No Response)

2. Is the manuscript technically sound, and do the data support the conclusions?

Reviewer #1: No

Reviewer #2: Partly

3. Has the statistical analysis been performed appropriately and rigorously? 

Reviewer #1: Yes

Reviewer #2: Yes

4. Have the authors made all data underlying the findings in their manuscript fully available?

Reviewer #1: Yes

Reviewer #2: Yes

5. Is the manuscript presented in an intelligible fashion and written in standard English?

Reviewer #1: Yes

Reviewer #2: Yes

6. Review Comments to the Author

Reviewer #1: The authors have responded to most of my concerns appropriately and they have softened the claims about this paradigm as a measure of episodic memory. However, I am still concerned about whether this is a measure of episodic memory as this increases the interest the paradigm might garner in the field.

As it stands I do not see convincing evidence of this. Certainly they have a subset of animals that show incidental learning which is interesting but I am not sure it is publishable as the experimental design stands now (longer traces would be helpful).

One approach might be to frame the paradigm within the idea that maybe all new one-trial memories are initially episodic and your incidental task captures this phenomenon (see Heterarchic reinstatement of long-term memory: A concept on hippocampal amnesia in rodent memory research. Lee JQ, Zelinski EL, McDonald RJ, Sutherland RJ. Neurosci Biobehav Rev. 2016 Dec;71:154-166.)

Reviewer #2: Unfortunately, the authors have not addressed my concerns. The entire manuscript revolves around episodic memory and both I and the other reviewer explicitly stated that there is no compelling evidence that this task relates to episodic memory. As indicated by the other reviewer, this is Pavlovian conditioning task.

The authors state "relevance" to episodic memory:

"A behavioral task does not have to model all aspects of episodic memory to be

useful in learning about episodic memory. This is why ‘episodic memory’ is often

mentioned in the text. However, we scrutinized the text and found some statements

that could be interpreted as presenting the OTTER to be an episodic memory task. We

added an explicit statement that we do not consider the OTTER to test true episodic

memory (lines 374-375):"

If it is not modelling episodic memory, then why is the entire manuscript developed around that topic?

Again, I think this is better suited as method paper for a new pavlovian task and not one within the episodic memory context.

7. PLOS authors have the option to publish the peer review history of their article (what does this mean?). If published, this will include your full peer review and any attached files.

Reviewer #1: No

Reviewer #2: No

---

## [Author Response · Author response to Decision Letter 1]

23 Mar 2023

Response to the reviewers

27th Feb 2023

We express our gratitude to the reviewers for providing us with constructive feedback. We concur with their assessment that the task primarily serves as a test of classical conditioning rather than episodic memory. To acknowledge this fact, we have revised the introduction section of the manuscript and incorporated several modifications to the discussion section. Yet we believe that the task, being incidental and limited to a single trial, does represent some fundamental components essential for establishing episodic-like memories. Nevertheless, the nature of the task does not allow us to conclude that it is a definitive episodic memory task. 

Reviewer #1: The authors have responded to most of my concerns appropriately and they have softened the claims about this paradigm as a measure of episodic memory. However, I am still concerned about whether this is a measure of episodic memory as this increases the interest the paradigm might garner in the field.

As it stands I do not see convincing evidence of this. Certainly they have a subset of animals that show incidental learning which is interesting but I am not sure it is publishable as the experimental design stands now (longer traces would be helpful).

Response to Reviewer 1: Thank you for your comment. We would like to clarify that our aim was to highlight that the OTTER task assesses several aspects that are relevant to episodic memory, but it is not designed as a specific test of episodic memory. There are many different tasks that explicitly target only some aspects of episodic memory, and they are still considered valuable in this line of research. Therefore, we believe that the OTTER task has a place among these tasks and can contribute to our understanding of the underlying mechanisms of episodic-like memory formation. We appreciate your feedback and hope that this clarification will address your concerns.

Reviewer 1: One approach might be to frame the paradigm within the idea that maybe all new one-trial memories are initially episodic and your incidental task captures this phenomenon (see Heterarchic reinstatement of long-term memory: A concept on hippocampal amnesia in rodent memory research. Lee JQ, Zelinski EL, McDonald RJ, Sutherland RJ. Neurosci Biobehav Rev. 2016 Dec;71:154-166.)

Response to Reviewer 1: Thank you for your helpful suggestion. We utilized the arguments presented in this publication to support the relevance of the OTTER task to the field of episodic memory research. Lines 37-40: “Finally, it has been proposed that one-trial memories are initially encoded as episodic, and that different memory systems might differ in their learning rates [12]. Our task featuring one-trial learning might prove helpful in testing this hypothesis.”

Reviewer #2: Unfortunately, the authors have not addressed my concerns. The entire manuscript revolves around episodic memory and both I and the other reviewer explicitly stated that there is no compelling evidence that this task relates to episodic memory. As indicated by the other reviewer, this is Pavlovian conditioning task.

The authors state "relevance" to episodic memory:

"A behavioral task does not have to model all aspects of episodic memory to be

useful in learning about episodic memory. This is why ‘episodic memory’ is often

mentioned in the text. However, we scrutinized the text and found some statements

that could be interpreted as presenting the OTTER to be an episodic memory task. We added an explicit statement that we do not consider the OTTER to test true episodic memory (lines 374-375)"

If it is not modeling episodic memory, then why is the entire manuscript developed around that topic?

Again, I think this is better suited as method paper for a new pavlovian task and not one within the episodic memory context.

Response to Reviewer 2: We apologize that our revisions did not meet your expectations, but we respectfully disagree with your assessment. 

Following the feedback received, we have made revisions to the paper to clarify that the OTTER task is not intended to model episodic memory, nor is it designed as an episodic-like memory task, as defined by the widely accepted "what-when-where" criteria. Despite this, we believe that the task remains a valuable and relevant tool for researchers in the field of animal research who seek to investigate the nature and dysfunction of human episodic memory.

We find the OTTER task relevant to episodic memory for several reasons, despite it not being an episodic-like memory task:

- Natural human episodic memory in the everyday ecological context has a ‘one-trial’ character: the events that are later recollected as an episodic memory occur only once (it would be logically incoherent to say that exactly the same event occurred twice, since the time would be different; thus another, albeit similar, event has occurred later),

- The human episodic memory is also incidental: at least some (and possibly most) of the episodic memories that a person acquires were not committed to memory intentionally, the person did not prepare for ‘a study session’ with the intent to remember the episode and all of the elements that are later recollected,

- The OTTER task is both one-trial and incidental in character.

These important aspects of human episodic memory are severely understudied in animal paradigms that aim to elucidate human episodic memory. In our opinion, the lack of research on these aspects of episodic memory is due to the lack of suitable behavioral memory-binding tasks with one-trial and incidental characteristics.

Another way to express why we consider the OTTER task to be relevant to episodic memory and of potential interest to the field even though it does not model episodic memory is illustrated by the following analogy:

 For example, an animal can never (by definition) be said to suffer from schizophrenia (the diagnostic criteria require the presence of hallucinations and delusions - both of which are personal experiences that can not be assessed in animals by a clinical interview). A similar situation applies to episodic memory, defined as a memory of subjective, personally experienced event. 

Thus, people model schizophrenia in animals so that animals display schizophrenia-like behavior. The same goes for episodic-like behavior. 

In schizophrenia, the reduction of the pre-pulse inhibition of the startle response (PPI) is well-known phenomenon that is present in patients and has been researched for decades. However, it is not among the diagnostic criteria (i.e. the definition of schizophrenia). It is questionable if modeling only this aspect in animals constitutes a schizophrenia model (at best, it would model symptom that is not even necessary for the patient to be diagnosed with schizophrenia). And studying healthy, undisrupted PPI definitely does not model schizophrenia. So if a researcher were to investigate the physiologically normal processes underlying PPI in a completely healthy animal, they would not model schizophrenia and elicit no schizophrenia-like behavior. But it would still be highly relevant to schizophrenia and of considerable interest to the field as clarifying the underlying processes of PPI could aid in deciphering changes occurring in schizophrenia patients. And it would be correct and even desirable for the field of schizophrenia research should the author write the paper with an eye on the topic of schizophrenia. And even more so, in a hypothetical case, if the field was not paying much attention to the PPI physiology and dysfunction due to the lack of suitable behavioral paradigms for animals (and PPI not being part of the diagnostic definition of schizophrenia).

As reasoned above, the human episodic memory likely has one-trial and incidental character, but episodic memory is not defined based on these characteristics. From the standpoint of prevailing definition (human episodic or animal episodic-like memory), the characteristics of being incidental and one-trial are non-essential, accidental - same as the reduction of PPI in schizophrenia patients is unnecessary for the diagnosis, yet still highly relevant for the field of schizophrenia research.

We hope that we showed clearly why we consider the OTTER task to be relevant to episodic memory even though it is not a what-when-where episodic memory-like task. We really appreciate your comment as it prompted us to think very critically about arguments supporting OTTER as a episodic memory task. We have reworked the paper based on the both reviewers’ feedback and now clearly state in the paper that the OTTER is not an episodic memory task. However, we still consider the task to be relevant to episodic memory and of potential interest to the field.

---

## [Decision Letter · Decision Letter 2]

8 May 2023

PONE-D-22-24003R2Incidental Temporal Binding in Rats: a Novel Behavioral TaskPLOS ONE

Dear Dr. Brozka

Thank you for submitting your manuscript to PLOS ONE. After careful consideration, we feel that it has merit but does not fully meet PLOS ONE’s publication criteria as it currently stands. Therefore, we invite you to submit a revised version of the manuscript that addresses the points raised during the review process.

Please address the two main critiques by Reviewer 2, and include consideration of the two publications they reference in their review.

We look forward to receiving your revised manuscript.

Kind regards,

Robert Sutherland, Ph.D

Academic Editor

PLOS ONE

Journal Requirements:

Reviewers' comments:

Reviewer's Responses to Questions

**Comments to the Author**

1. If the authors have adequately addressed your comments raised in a previous round of review and you feel that this manuscript is now acceptable for publication, you may indicate that here to bypass the “Comments to the Author” section, enter your conflict of interest statement in the “Confidential to Editor” section, and submit your "Accept" recommendation.

Reviewer #1: All comments have been addressed

Reviewer #2: (No Response)

2. Is the manuscript technically sound, and do the data support the conclusions?

Reviewer #1: Yes

Reviewer #2: Partly

3. Has the statistical analysis been performed appropriately and rigorously? 

Reviewer #1: Yes

Reviewer #2: Yes

4. Have the authors made all data underlying the findings in their manuscript fully available?

Reviewer #1: No

Reviewer #2: Yes

5. Is the manuscript presented in an intelligible fashion and written in standard English?

Reviewer #1: Yes

Reviewer #2: Yes

6. Review Comments to the Author

Reviewer #1: (No Response)

Reviewer #2: This is an interesting task and could be of some use. Unfortunately, I still have issues related to the contextualization around episodic memory. I still do not understand how this task is different than other one-trial tasks.

The incidental argument, in my view, is poorly presented. The authors, in the first paragraph of the introduction, define incidental as "memory that can be tested when subjects are unaware that they will be tested on recall". When would rats be aware that they will be tested for memory? Basically, the authors' operational definition for rodent memory is inadequate.

Similarly, the first sentence of the second paragraph emphasizes the need to study time gaps to better understand episodic memory, which again I do not find supported by the literature on episodic memory. The importance, within the episodic memory literature, is the time stamp between events (the sequence of two different events). This task assesses a CS-US association and not a time stamp or sequence of events. Moreover, the authors do not study time gaps in this manuscript as they only examined one interval between the CS and US.

I stand firm that the introduction should revolve around a new one-trial conditioning task. The episodic memory component should be reduced.

The discussion is improved, but the "declarative" argument in the discussion should be revised as it is much too strong and overstated for this task. Declarative memory implies describing facts and events (use of language). The OTTER task does not provide the richness of detail associated with declaration. Further the authors should briefly discuss the view that rodents may not have episodic memory (see Zenthal, 2013).

I strongly recommend that the authors read and come to cite these two papers:

Pause, B. M., Zlomuzica, A., Kinugawa, K., Mariani, J., Pietrowsky, R., & Dere, E. (2013). Perspectives on episodic-like and episodic memory. Front Behav Neurosci, 7, 33. doi: 10.3389/fnbeh.2013.00033

Zentall, T. R. (2013). Animals Represent the past and the Future. Evolutionary Psychology, 11(3), 147470491301100307. doi: 10.1177/147470491301100307

7. PLOS authors have the option to publish the peer review history of their article (what does this mean?). If published, this will include your full peer review and any attached files.

Reviewer #1: **Yes: **Dr. Robert J. McDonald

Reviewer #2: No

---

## [Author Response · Author response to Decision Letter 2]

18 May 2023

Reviewer #2: This is an interesting task and could be of some use. Unfortunately, I still have issues related to the contextualization around episodic memory. I still do not understand how this task is different than other one-trial tasks.

Radostova et al.: We greatly appreciate your partial approval of the manuscript. We would like to clarify that the OTTER task is not claimed to be entirely distinct but expands the range of existing one-trial tasks. What we find particularly original about the OTTER task is its utilization of two types of motivation to influence animal behavior.

Reviewer #2: The incidental argument, in my view, is poorly presented. The authors, in the first paragraph of the introduction, define incidental as "memory that can be tested when subjects are unaware that they will be tested on recall". When would rats be aware that they will be tested for memory? Basically, the authors' operational definition for rodent memory is inadequate.

Radostova et al.: We apologize for not fully explaining this manuscript part. We added a following section: “To ensure that memory encoding is incidental, the memory test should be unexpected when the to-be-remembered event occurs [Zentall 2013]. Repeated test trials can lead animals to anticipate future memory testing. This repetition helps animals learn the task rules and identify relevant information, making them more motivated to encode the information intentionally”.

Hopefully, this clarifies the issue.

Reviewer #2: Similarly, the first sentence of the second paragraph emphasizes the need to study time gaps to better understand episodic memory, which again I do not find supported by the literature on episodic memory. The importance, within the episodic memory literature, is the time stamp between events (the sequence of two different events). This task assesses a CS-US association and not a time stamp or sequence of events. Moreover, the authors do not study time gaps in this manuscript as they only examined one interval between the CS and US.

Radostova et al.: We changed the term ‘episodic memories’ to ‘complex memories.’ We think this minor change resolves your concerns about the paragraph.

Reviewer #2: I stand firm that the introduction should revolve around a new one-trial conditioning task. The episodic memory component should be reduced. 

Radostova et al.: We did as you requested: we erased all mentions of the ‘episodic’ memory (but one, requested by the other reviewer) from the introduction paragraph. Thank you for this suggestion, as we think addressing all complex memories is an added value.

Reviewer #2: The discussion is improved, but the "declarative" argument in the discussion should be revised as it is much too strong and overstated for this task. Declarative memory implies describing facts and events (use of language). The OTTER task does not provide the richness of detail associated with declaration. 

Radostova et al.: We appreciate your perspective on the matter. We have replaced the term "declaring" with "showing" and the phrase "declarative behavior" with "response." Thank you for your valuable comment. We believe the paragraph now conveys the intended meaning more appropriately.

Reviewer #2: Further the authors should briefly discuss the view that rodents may not have episodic memory (see Zenthal, 2013).

Radostova et al.: Thank you for your feedback. Considering your suggestion, we have removed all statements related to episodic memory as advised. Therefore, there is no need further to discuss the presence of episodic memory in rodents. However, we did cite Zentall 2013 in the introduction when discussing ‘unexpected question’ tasks.

Reviewer #2: I strongly recommend that the authors read and come to cite these two papers:

● Pause, B. M., Zlomuzica, A., Kinugawa, K., Mariani, J., Pietrowsky, R., & Dere, E. (2013). Perspectives on episodic-like and episodic memory. Front Behav Neurosci, 7, 33. doi: 10.3389/fnbeh.2013.00033

● Zentall, T. R. (2013). Animals Represent the past and the Future. Evolutionary Psychology, 11(3), 147470491301100307. doi: 10.1177/147470491301100307

Radostova et al.: We appreciate your valuable recommendations and find the suggested reads highly interesting. However, since we have significantly reduced references to episodic memory in our discussion, citing these sources is no longer relevant.

---

## [Editor Report · Decision Letter 3]

2 Jun 2023

Incidental Temporal Binding in Rats: a Novel Behavioral Task

PONE-D-22-24003R3

Dear Dr. Hana Brozka,

We’re pleased to inform you that your manuscript has been judged scientifically suitable for publication and will be formally accepted for publication once it meets all outstanding technical requirements.

Kind regards,

Robert Sutherland, Ph.D

Academic Editor

PLOS ONE
---

## [Editor Report · Acceptance letter]

13 Jun 2023

PONE-D-22-24003R3 

Incidental Temporal Binding in Rats: a Novel Behavioral Task. 

Dear Dr. Brozka:

I'm pleased to inform you that your manuscript has been deemed suitable for publication in PLOS ONE. Congratulations! Your manuscript is now with our production department. 

Kind regards, 

on behalf of

Dr. Robert Sutherland 

Academic Editor

PLOS ONE